# A Review of Lignan Metabolism, Milk Enterolactone Concentration, and Antioxidant Status of Dairy Cows Fed Flaxseed

**DOI:** 10.3390/molecules24010041

**Published:** 2018-12-22

**Authors:** André F. Brito, Yu Zang

**Affiliations:** Department of Agriculture, Nutrition, and Food Systems, University of New Hampshire, Durham, NH 03824, USA; yz1040@wildcats.unh.edu

**Keywords:** animal health, cattle, enterolignan, human health, pharmacokinetic, ruminant, secoisolariciresinol diglucoside

## Abstract

Lignans are polyphenolic compounds with a wide spectrum of biological functions including antioxidant, anti-inflammatory, and anticarcinogenic activities, therefore, there is an increasing interest in promoting the inclusion of lignan-rich foods in humans’ diets. Flaxseed is the richest source of the lignan secoisolariciresinol diglucoside—a compound found in the outer fibrous-containing layers of flax. The rumen appears to be the major site for the conversion of secoisolariciresinol diglucoside to the enterolignans enterodiol and enterolactone, but only enterolactone has been detected in milk of dairy cows fed flaxseed products (whole seeds, hulls, meal). However, there is limited information regarding the ruminal microbiota species involved in the metabolism of secoisolariciresinol diglucoside. Likewise, little is known about how dietary manipulation such as varying the nonstructural carbohydrate profile of rations affects milk enterolactone in dairy cows. Our review covers the gastrointestinal tract metabolism of lignans in humans and animals and presents an in-depth assessment of research that have investigated the impacts of flaxseed products on milk enterolactone concentration and animal health. It also addresses the pharmacokinetics of enterolactone consumed through milk, which may have implications to ruminants and humans’ health.

## 1. Introduction

Lignans are polyphenolic, phytoestrogenic compounds known to display a wide range of biological functions, including weak estrogenic and cardioprotective activities, as well as antiestrogenic, antioxidant, anti-inflammatory, and anticarcinogenic properties [1,2,3]. The weak and antiestrogenic effects of lignans are caused by distinct transactivation activities of estrogen receptors between the enterolignans enterodiol (ED) and enterolactone (EL) [4]. There is a growing interest in promoting the consumption of lignan-rich foods because of the potential benefits to human health. The outer fibrous-containing layers of flaxseed (*Linum usitatissimum* L.) is the richest source of the lignan secoisolariciresinol diglucoside (SDG) [5], which accounts for over 95% of the total lignans found in flax [6]. In ruminants, the rumen appears to be the main site for conversion of SDG into the mammalian lignans ED and EL [7,8,9,10]. However, only EL was detected in milk of dairy cows fed flaxseed meal (FM) [11] possibly because of ruminal dehydrogenation reactions converting ED to EL like those occurring in humans [12]. This suggests that EL-enriched milk can be used as a source of lignans for humans due to the following reasons: (1) milk is consumed by a large part of the world population despite regional differences in per capita consumption [13], (2) global consumption of milk is projected to increase by 60% between 2005/2007 and 2050, particularly in regions where the population traditionally consumes less milk such as East and North Africa, sub-Saharan Africa, and South and East Asia [14], and (3) a poor and variable consumption of plant lignans worldwide [15].

Hulls, meal, and whole seeds are flaxseed products that have been used as sources of the lignan SDG to improve the concentration of EL in milk of dairy cows [11,16,17,18,19]. It is important to note that other ingredients (e.g., forages, cereal grains, protein supplements) used in diets of dairy cows also provide lignans. Therefore, comparison of milk EL concentrations across experiments should consider the contribution of lignans from non-flaxseed feedstuffs. Diets containing sources of nonstructural carbohydrates (NSC) with different ruminal degradability (e.g., ground corn vs. liquid molasses) also have been shown to affect the EL concentration in milk of dairy cows fed FM [18]. Despite the growing knowledge regarding the impact of flaxseed supplementation on milk EL concentration in the last 10 years, little is known about how dietary manipulation affects the ruminal microbiome and EL production in dairy cows. Research in this area is needed to unravel dietary strategies suitable to modulate the concentration of EL in dairy cows’ milk.

In addition to human health benefits, flaxseed lignans can be also used as natural antioxidants to improve animal health via upregulation of antioxidant enzymes. Newborn dairy calves and periparturient dairy cows are prone to oxidative stress and immune depression [20,21]. Previous research revealed that the antioxidant activity of plant enterolignans is stronger than that of vitamin E [22]. Furthermore, weanling albino rats receiving 10% flaxseed (1.5 g/kg of body weight) during 14 d followed by a challenge with a toxin (i.e., carbon tetrachloride) known to downregulate the hepatic expression of antioxidant enzymes were able to restore the activities of superoxide dismutase (SOD), catalase (CAT), and glutathione peroxidase (GPx) by 95, 182, and 136%, respectively, compared with the control treatment [23]. Altogether, these results are encouraging and open new opportunities to explore the use of flaxseed products or flaxseed-derived lignans as bioactive sources to mitigate oxidative stress in newborn, growing, and adult dairy cattle. 

The primary objective of this review is to present an in-depth summary and evaluation of research that have investigated the impacts of flaxseed hulls (FH), FM, and whole seeds flaxseed (WF) on milk EL concentration and animal health. We also covered the metabolism of lignans in the gastrointestinal tract of humans and animals and the pharmacokinetics of milk EL consumed by newborn dairy calves, which may have implications to ruminants and humans’ health.

## 2. Metabolism of Lignans in the Gastrointestinal Tract

The flaxseed lignans SDG, secoisolariciresinol (SECO), pinoresinol, lariciresinol, and matairesinol are converted by the gut microbiota of humans [6,24] and ruminants [7,8,9,10] to the enterolignans ED and EL. In contrast, the lignan isolariciresinol, also derived from flaxseed, is not converted to ED and EL [25]. Enterodiol and EL are named mammalian lignans or enterolignans because they are produced in the gut of humans and other mammals and not found in plant tissues [26]. A simplified pathway highlighting the conversion of plant lignans to enterolignans in humans is presented in Figure 1. Consortia of gut microorganisms appear to be involved in the sequential catalytic reactions reported in Figure 1, including 28 bacterial species belonging to 12 different genera such as *Bacteroides*, *Clostridium*, *Bifidobacterium*, and *Ruminococcus* among others according to previous research [12,27,28,29,30,31,32,33]. After conversion of lignans into ED and EL, these enterolignans are absorbed in the large intestine followed by conjugation as glucuronides and sulfates based on in vitro work using human colon epithelial cells [34]. Conjugated EL and ED undergo extensive first-pass metabolism and enterohepatic recirculation [34,35], as well as deconjugation by colonic bacterial β-glucuronidases and sulfatases followed by reabsorption [36]. It has also been shown that conjugation of EL takes place not only in the colon, but also in the small intestine and liver microsomes of humans and rats according to in vitro enzymatic kinetic analysis of EL glucuronidation [35].

An investigation of the relationship among the gut microbial community, urinary EL excretion, and diet from a 3-d food record of 115 premenopausal American women (40–45 years old) revealed a significant positive association between EL excretion and either the gut microbial community or its diversity [37]. They also demonstrated that the gut microbial community associated with high EL production was distinct and enriched in *Moryella*, *Acetanaerobacterium*, *Fastidiosipila* spp., and *Streptobacillus* spp. [37]. Interestingly, these 4 bacterial genera were not part of those typically related to the sequential pathway of lignans catabolism [12,27,28,29,30,31,32,33]. However, despite these genera not being previously linked to EL production, they are closely related to those involved in the metabolism of lignans [37]. Recently, the complete metabolic pathway of pinoresinol and lariciresinol was unraveled using comparative genomics and transcriptional profiling (RNAseq) prepared from stool samples, thus indicating that the conversion of dietary lignans to bioactive enterolignans is a common route adopted by the gut microbiota of humans [38]. These results are an important step for advancing the molecular genetic understanding of the gut bioactivation of lignans and other plant secondary compounds to downstream metabolites relevant to humans’ health [38].

In ruminants, it is conceivable that deglycosylation, demethylation, dehydroxylation, and dehydrogenation reactions like those reported in humans (Figure 1) are also involved in the metabolism of lignans, but little is known about which ruminal bacteria species or consortia participate in these reactions. Lignans present in FH and WF were both converted to mammalian lignans by ruminal and fecal microbiota of dairy cows during in vitro incubations [7]. While EL was the major enterolignan produced by the ruminal microbiota, the fecal counterpart yielded primarily ED [7]. In a study conducted using ruminally-cannulated goats, the concentrations of SDG, ED, and EL increased significantly in both rumen and serum following ruminal infusion of SDG (1 mg/kg of body weight) [9]. These authors also observed that the ruminal and serum concentrations of EL were approximately 2-fold greater than those of ED [9], indicating that EL is the predominant enterolignan in the rumen, which agree with results from another study [7]. The role of the ruminal microbiota and the effects of flaxseed oil (FO) in the metabolism of flaxseed-lignans and concentrations of EL in biological fluids have been also investigated [8]. Flaxseed oil is a rich source of polyunsaturated fatty acids (PUFA) [39], which are known to be toxic for certain species of ruminal microorganisms [40,41]. Therefore, feeding sources rich in PUFA may interfere with the ruminal metabolism of flaxseed-lignans and ultimately affect the concentrations of EL in biological fluids. The concentrations of EL increased by an average of 1,755% in urine, 238% in plasma, and 925% in milk of cows administered with FH in the rumen compared with FO and FH infused in the abomasum [8]. However, no significant differences in the concentrations of EL in urine, plasma, and milk were observed when FO was administered in the rumen and FH infused in the abomasum [8], which confirm that rumen is the major site for conversion of SDG to EL. In addition, the ruminal concentration of EL increased linearly and a strong correlation (r = 0.76) between EL concentrations in ruminal fluid and milk was observed in dairy cows fed incremental amounts of FM [0, 5, 10, and 15% of the diet dry matter (DM)] [10,42], further reinforcing the key role of the ruminal microbiota in the metabolism of flaxseed-SDG.

It appears that in ruminants, ED and EL are absorbed in the rumen and intestines [10,43,44], possibly as conjugated forms like other phytoestrogens including formononetin, daidzein, and equol [43]. Interestingly, sheep had a greater conjugative activity than cattle in most parts of the gastrointestinal tract evaluated (i.e., rumen, reticulum, omasum) except in the small intestine [43]. In humans, deconjugation performed by gut microbial β-glucuronidases and sulfatases is known to enhance the reabsorption of ED and EL [36,45,46]. Studies conducted with lactating dairy cows showed no relationship between flaxseed supplementation (FH or FM) and activity of microbial β-glucuronidase in the rumen [8,10,47], thus suggesting that this enzyme has little or no involvement in the ruminal absorption of EL, possibly because conjugation occurs during or after cell uptake of enterolignans [43]. In fact, when the ruminal activity of microbial β-glucuronidase decreased in dairy cows fed FH [48], the concentrations of EL in rumen, plasma, urine, and milk increased compared with the control diet. However, additional research is needed to elucidate the actual mechanisms involved in the absorption of enterolignans in ruminant animals. Likewise, research investigating the potential effects of intestinal β-glucuronidases on deconjugation of enterolignans before reabsorption in the large intestine of ruminants is warranted.

Studies in which oil (FO or sunflower) was administered in the rumen or infused in the abomasum also helped to shed light on the gastrointestinal tract metabolism of lignans in dairy cows. Oil sources rich in n-3 PUFA such as FO are known to inhibit the growth of ruminal microorganisms involved in fiber degradation (e.g., *Butyrivibrio*, *Ruminococcus*) and methanogenesis (e.g., *Methanobrevibacter*) [40,41]. β-glucuronidase activity in humans has been attributed to colonic bacteria belonging to the genera *Ruminococcus*, *Bacteroides*, *Bifidobacterium*, and *Eubacterium* [49], which are also found in the rumen [50,51]. Thus, it is conceivable that FO may inhibit ruminal bacteria with β-glucuronidase activity. In fact, FO reduced microbial β-glucuronidase activity when it was administered in the rumen, but not during abomasal infusion in lactating dairy cows [8]. These results [8] imply that ruminal bacteria with predominant β-glucuronidase activity may be more susceptible to the toxic effects of FO than those primarily involved in the conversion of SDG to EL as the concentration of EL in the rumen was not affected by the site of FO supplementation (rumen or abomasum). Compared with the control treatment, fecal β-glucuronidase activity tended to increase in dairy cows fed FH and no change was detected with abomasal infusion of FO in another experiment [48]. In contrast, it was found that feeding FM and infusing sunflower oil (n-6 PUFA source) in the abomasum of lactating dairy cows decreased fecal β-glucuronidase activity relative to the control treatment [47]. It has been shown that the ruminal microbiota can be modulated by modifying the dietary PUFA profile and similar processes may also take place in the large intestine of ruminants, which may explain to a certain extent these inconsistent results in fecal β-glucuronidase activity [8,47,48]. Changes (increase or decrease) in 16S rRNA copy numbers of ruminal microorganisms such as *Butyrivibrio*, ciliate protozoa, methanogens, *Selenomonas ruminantium*, and *Streptococcus bovis* were detected during an in vitro rumen simulation technique study in which fermenters were dosed with diets rich in n-6 PUFA (i.e., sunflower oil) or a n-6/n-3 PUFA mix (i.e., sunflower oil plus fish and algae oil) [52]. Overall, ruminal or fecal microbiota β-glucuronidase activity appears to have limited biological importance for the absorption of EL in lactating dairy cows fed different flaxseed products or abomasally-infused with n-3 or n-6 PUFA oil sources. 

As mentioned earlier, there is scarce information about the role of ruminal microbiota species in the metabolism of plant-derived lignans. Ruminal supplementation of SDG stimulated the growth of the bacterium *Ruminococcus gnavus*, which is likely involved with glucuronidase activity in the rumen [9]. In fact, *R. gnavus* E1, an anaerobic bacterium belonging to the dominant human gut microbiota, expresses the gene *gnus* known to encode for the β-glucuronidase enzyme [49]. In a more recent study, the concentration of total ruminal bacteria 16S rRNA obtained using qPCR did not differ in cows fed incremental amounts of FM [42]. However, additional PCR-DGGE and DNA extraction analyses using bands from cows fed 15% FM showed that several genera contributed to the metabolism of lignans, particularly *Prevotella* spp. [42]. Moreover, a follow-up in vitro pure culture assay revealed that 11 ruminal bacteria species were able to metabolize SDG to SECO, with bacteria from the genus *Prevotella* being the most efficient followed by *Butyrivibrio fibrisolvens* and *Peptostreptococcus anaerobius*, whereas *Ruminococcus albus*, *Eubacterium ruminantium*, *Butyrivibrio proteoclasticus*, and *Ruminococcus flavefaciens* showed the least conversion efficiency [42]. Their data also suggested that intermediate compounds between the SDG to EL pathway were formed during in vitro pure culture incubations due to the presence of unidentified peaks in the chromatograms [42]. Overall, the genus *Prevotella* appears to be the most relevant in the metabolism of plant lignans to enterolignans in ruminants. However, the current knowledge regarding ruminal microbiota diversity and function in young and adult ruminants fed different sources of flaxseed is limited and warrants further research.

## 3. Effects of Flaxseed Products on Milk EL Concentration

Table 1 summarizes results from 15 studies in which milk EL concentration was measured in dairy cows fed different flaxseed products (i.e., FH, FM, WF) and NSC sources. The nutritional profile of flaxseed products used in studies summarized in Table 1 are presented in Table 2. 

### 3.1. Dose-Response Studies and Milk EL Concentration

Four dose-response studies using FH (1 experiment) [17], FM (2 experiments) [10,11], and WF (1 experiment) [53] have been conducted to date (Table 1). In three out of four experiments, the milk concentration of EL increased linearly in response to incremental amounts (diet DM basis) of FH (0, 5, 10, and 20%) or FM (0, 5, 10, and 15%). Compared with the control diet, feeding 20% FH increased the concentration of milk EL by approximately 250% [17]. The milk concentrations of EL increased by approximately 110% [11] and 330% [10] relative to control treatments when cows were fed the greatest amount of FM (i.e., 15%). In contrast, only a positive linear trend in milk EL concentration was observed in response to increasing amounts of WF (0, 5, 10, and 15%) [53]. Flaxseed hulls (mean = 1% SDG) and FM (mean = 1.6% SDG) contain greater concentrations of SDG than WF (mean = 0.6% SDG; see Table 2), thus consistent with a more pronounced response in milk EL concentration with feeding FH or FM versus WF. No curvilinear responses were detected in these four dose-response studies, indicating that a theoretical maximum concentration of milk EL was not achieved in diets containing up to 15% FM, 15% WF, or 20% FH. These results also suggest that ruminal and intestinal absorptive mechanisms were not saturated by increased concentrations of EL. However, there are limitations regarding the amount of flaxseed products that can be included in dairy diets due to environmental and milk production concerns associated with excess intake of crude protein or crude fat depending on the flax source used. As shown in Table 2, FM is a protein supplement (mean = 37.2% crude protein), while FH can be used as both lipid (mean = 28.4% crude fat) and protein sources (mean = 22.4% crude protein); likewise, WF contains high concentration of lipids (mean = 34.9% crude fat) and moderate crude protein content (mean = 23.5%).

High intake of crude protein can lead to excess N excretion to the environment and poor N use efficiency in lactating dairy cows [59,60]. Excess consumption of fat (>5% of the diet DM) has been associated with depressed DM intake, milk production, and ruminal fiber digestibility [59].

### 3.2. Comparison of Flaxseed Products and Animal Variation in Milk EL Concentration

We are aware of only one publication that compared, in the same experiment, the effect of flaxseed products on milk EL concentration in dairy cows (i.e., [16]; see Table 1). In this study [16], 24 lactating dairy cows were used in a randomized complete block design in which animals were assigned to a control diet without flaxseed supplementation or 10% of the diet DM as FM or WF. It was observed that relative to the control treatment, the milk concentration of EL increased by an average of 178% in cows fed FM or WF. However, no differences in the concentration of milk EL was found between FM and WF. Even though milk EL yield (mg/d) did not differ with feeding FM versus WF, only cows supplemented with FM had a significant increase in milk EL output (+216%) compared with the control animals. For the remaining studies summarized in Table 1, including the dose-response experiments (discussed above) and the feeding trials that evaluated different NSC sources and FM supplementation (discussed next section), milk EL concentration improved in all except one study (i.e., [47]). In their experiment [47], 8 ruminally-cannulated dairy cows were used in a replicated 4 × 4 Latin square design with a 2 × 2 factorial arrangement of treatments. The concentrations of milk EL averaged 75 and 122 n*M* in cows fed diets without and with FM supplementation, respectively. Despite an average increase of 63% in milk EL concentration comparing FM- versus non-FM diets [47], this difference did not reach statistical significance possibly because of the low number of animals used and the large cow-to-cow variability in milk EL content. For instance, the 95% confidence interval for milk EL concentration ranged from 32 to 161 n*M* (control), 35 to 175 n*M* (250 g/d abomasal infusion of sunflower oil), 46 to 221 m*M* (13.7% FM), and 63 to 312 n*M* (13.7% FM plus 250 g/d abomasal infusion of sunflower oil) [47]. 

A large interindividual variation in the concentration of the phytoestrogen equol in milk of dairy cows has been reported, with values ranging from 400 to 2,600 µg/kg across treatments in two experiments [61]. Similarly, we [19] observed a large interindividual variation in milk EL yield in dairy cows fed varying levels of NSC sources and 15% FM (see Figure 2), which is consistent with previous research [61]. This large cow-to-cow variability cannot be entirely explained by differences in dietary composition or phytoestrogens intake so that other factors such as ruminal microbiota profile, digesta passage rate, and dairy cattle genetics may be also involved [61]; however, the actual biological mechanisms underpinning this wide interindividual variability are not well understood. Previous researchers reported that EL is a transported substrate and likely a competitive inhibitor of the ATP-binding cassette subfamily G2 (ABCG2) protein [62], which is known to transport phytoestrogens and their conjugated metabolites [63,64,65]. It was further demonstrated that the milk-to-plasma ratio of EL decreased significantly in the Abcg2^(−/−)^ knockout female mice phenotype compared with the wild-type group (0.4 vs. 6.4) [62]. A subsequent study showed that EL was used as substrate to the bovine ABCG2 variant Y in vitro and was also actively secreted in milk resulting in a 2-fold increase in its milk-to-plasma ratio in Y/S heterozygous versus Y/Y homozygous cows [66]. The bovine ABCG2 Y581S variant has been described as a gain-of-function polymorphism that increases milk secretion and decreases plasma levels of its substrates [67,68,69]. Taken together, the ABCG2 protein and its variant Y581S appear to contribute to the interindividual variation of EL secretion in milk of dairy cows opening the possibility for controlling, through genetic selection or other management tools, the amount of enterolignans consumed by the population [61]. Improved knowledge of lignans metabolism in ruminants is needed because high intake of phytoestrogens may result in adverse health effects, particularly in critical stages of infant development [70,71] and during lactation and pregnancy [72]. Therefore, timing of exposure to phytoestrogens is key for capitalizing on health benefits while minimizing undesirable health outcomes [73]. In a recent literature review, the authors stated that current evidences regarding the potential health benefits of phytoestrogens are not so convincing that clearly outweigh the possible health risks (e.g., decreased fertility, increased risk of cancer in estrogen-sensitive tissues) [74]. They concluded that data currently available are not sufficient to support a more refined (semi) quantitative risk–benefit analysis, implying that a definite conclusion on potential health benefit outcomes of phytoestrogens cannot be made [74].

### 3.3. Impact of NSC Sources and FM on Milk EL Concentration

To the best of our knowledge, only three studies have investigated the impact of different NSC sources on milk EL concentration in dairy cows fed FM (see Table 1). It is well established that in relation to starch, sugars are more rapidly fermented in the rumen [75], implying that NSC sources with different degradability in the rumen may change ruminal fermentation processes, digesta passage rate, and microbiota growth and species composition. Compared with ground corn, liquid or dried molasses has greater concentration of sucrose [18,76]. The effects of supplemental NSC (ground corn vs. liquid molasses) and rumen-degradable protein (soybean meal-sunflower meal mix vs. FM) on milk EL concentration have been evaluated in dairy cows fed grass hay-based diets [18]. No significant rumen-degradable protein by NSC source interaction was observed for milk EL concentration. However, significant rumen-degradable protein and NSC source effects were detected; cows fed diets containing (DM basis) 16% FM and 12% liquid molasses had 288 and 53% more EL in milk than those fed rations consisting of 16% soybean meal-sunflower meal mix and 12% ground corn, respectively. Therefore, liquid molasses may select for ruminal microorganisms with better capacity to convert FM-SDG to EL than ground corn [18]. A follow-up study evaluated the effects of replacing ground corn with incremental amounts of liquid molasses (0, 4, 8, and 12% of the diet DM) on milk EL concentration in dairy cows fed 15% FM [19]. It was hypothesized that the concentration of EL in milk would be modulated by possible changes in DM intake (also affecting SDG intake) when varying the dietary proportions of liquid molasses and ground corn. Only a cubic trend was observed for milk EL concentration despite the linear decrease in SDG intake with replacing ground corn by liquid molasses [19]. Although this cubic trend is difficult to explain biologically, the lack of a precursor-product relationship suggests that the ruminal output of EL seems to be more affected by the microbiota metabolism of SDG than by SDG supply. Milk EL yield did not differ and averaged 1.38, 1.61, 1.36, and 1.52 mg/d in diets containing 0, 4, 8, or 12% liquid molasses, respectively [19]. *Prevotella* spp. have been reported to be one of the main converters of SDG to SECO, a lignan-derived metabolite that is further metabolized to ED and EL, presumably by additional ruminal microbiota species [42]. *Prevotella* species are also capable of utilizing starch, other non-cellulosic polysaccharides, and simple sugars as energy sources, yielding succinate as the major end-product of ruminal fermentation [77]. Therefore, it was not entirely surprising to obtain a curvilinear response for milk EL concentration with feeding various dietary levels of liquid molasses [19] because *Prevotella* spp. can utilize both starch and sugars [77]. 

Our laboratory conducted a third study evaluating the effect of sucrose and FO on milk EL concentration of dairy cows fed 15% FM [58]. Specifically, 16 lactating dairy cows were used in a replicated 4 × 4 Latin square design with the following arrangement of treatments (% of diet DM): (1) 8% soybean meal (control); (2) 5% sucrose + 15% FM; (3) 3% FO + 15% FM; and (4) 5% sucrose + 3% FO + 15% FM. As discussed above, *Prevotella* spp. have been shown to be involved in the metabolism of SDG [42] and NSC [77]. In addition, the genus *Prevotella* dominated the ruminal bacterial community when steers were fed diets containing 4% FO, suggesting that *Prevotella* species are possibly involved in the metabolism of PUFA [78]. We hypothesized [58] that sucrose and FO could synergistically interact to increase the concentration of EL in milk as sugars [77] and FO [78] have been shown to promote growth of *Prevotella* spp. Compared with the control diet (mean = 76.8 n*M* of milk EL), the average concentration of EL in milk increased 4-fold in cows fed 15% FM (mean = 321 n*M*). However, no differences in milk EL concentration was observed among the treatments containing FM supplemented with sucrose or FO or both [58]. Overall, our data [18,19,58] indicate that the use of NSC sources with different ruminal degradability did not consistently improve milk EL concentration. Differences in DM intake, milk production, type of forage, and forage-to-concentrate ratio may have contributed to the inconsistent results in milk EL content across our studies.

### 3.4. Dairy Breed and Milk EL Concentration

Holstein cows were used in all studies presented in Table 1 except in two experiments where Jerseys were selected [18,19]. A large interindividual variation for the milk concentration of equol has been reported, but this variability was more pronounced in Swedish Red than Norwegian Red dairy cows [61]. These results suggest that dairy cattle genetics may influence the output of phytoestrogens in milk. It is well known that Jersey cows produce milk with greater concentrations of fat and protein than Holsteins (e.g., [79]). However, we are not aware of any publication that has simultaneously compared Holstein versus Jersey cows in terms of milk EL concentration and yield. Therefore, data from [10,11,19,58] were used to make inferences regarding the concentration of milk EL between breeds. In these four studies cows received 15% FM in at least one dietary treatment (see Table 1 for details). The concentration of milk EL averaged 259 n*M* in Jerseys [19], and 78 n*M* [11], 650 n*M* [10], and 321 n*M* [58] in Holsteins. Compared with one study using Holsteins [11], the concentration and yield of milk EL in Jerseys increased by 3.3- and 2.8-fold, respectively [19]. Contrarily, the concentration and yield of milk EL were greater in two other studies with Holsteins [10,58] than Jerseys [19], suggesting that no conclusive relationship between dairy breed and milk EL could be established. It is important to note that this exercise is a gross evaluation of the potential effect of dairy breed on milk EL concentration so that any association between breed and milk EL should be done cautiously. Nevertheless, the mean concentration of milk EL ranged from 78 to 650 n*M* implying that genetics, dietary composition, and even analytical methods may be involved in this variation in milk EL across experiments [10,11,19,58]. For instance, a chromatographic method (i.e., HPLC) was used in one (i.e., [11]) of the four studies resulting in the lowest milk EL content (i.e., 78 n*M*). The concentration of EL in the remaining three studies [10,19,58] were analyzed colorimetrically using a commercial competitive enzymatic immunoassay, which led to an average milk EL concentration 425% greater than that obtained with HPLC [11]. Moreover, the ingredient composition of the basal diet, forage-to-concentrate ratio, and forage source may have changed the ruminal environment among these four studies ultimately impacting the concentration of EL in milk. Plant lignans such as matairesinol, pinoresinol, and lariciresinol are also converted to enterolignans [6,24,80], with pinoresinol and lariciresinol present in greater concentrations than SDG and matairesinol in several plant species [81]. Thus, it is conceivable that dietary ingredients other than flaxseed may also supply lignans to the ruminal microbiota, which can contribute to variation in milk EL concentration reported in the literature. 

## 4. Pharmacokinetics of Milk EL and Potential Implications on Animal and Human Health

Elevated blood concentrations of ED and EL have been associated with reduced risk of coronary diseases and colorectal adenoma in humans [82,83,84]. A dose-response relationship between flaxseed intake and serum concentrations of ED or EL was observed in a study conducted with healthy young women [85]. Moreover, a 5-fold increase in the urinary excretion of EL was found in rats fed pure EL compared with those fed plant lignans [86]. These authors [86] hypothesized that EL may be passively absorbed along the intestinal tract, while plant lignans must be first converted to EL by colonic microorganisms followed by absorption in a limited segment of the gut. A large interindividual variation in the blood concentration of enterolignans has been observed in humans, thus revealing differences in the capacity of the colonic microbiota in converting plant lignans to ED and EL [46,85,87]. Therefore, EL-enriched milk has potential to be used as an enterolignan source for improving human health, particularly because EL appears to be more bioavailable than plant lignans [86]. Periparturient dairy cows, as well as newborn and nursing dairy calves could also benefit from the antioxidant properties of EL due to their susceptibility to oxidative stress and depressed immune system [20,21]. However, there is limited information regarding the pharmacokinetics of EL derived from milk and we are not aware of any published research that have instigated the effects on EL-enriched milk on human or animal health.

Recently, we investigated the pharmacokinetics of EL in newborn dairy calves fed milk replacer or EL-enriched milk [58]. In newborn calves, suckling stimulates the reflex closure of the esophageal groove so that ingested milk or milk replacer bypass the reticulo-rumen down to the abomasum [50]. Thus, calves may be used as a translational model to make inferences about the pharmacokinetics of EL in simple-stomach mammals including humans. We hypothesized that the area under the curve and plasma concentration of EL would be greater in Holstein calves fed a single bolus of EL-enriched milk versus milk replacer [58]. The EL-enriched milk was collected from a Jersey cow fed 15% FM. On d 5 of life, 20 calves (10 males and 10 females) were administered 2 L of milk replacer (low-EL treatment: 123 n*M* of EL) or 2 L of EL-enriched milk (high-EL treatment: 481 n*M* of EL) during the morning feeding. The area under the curve for the plasma concentration of EL, which was determined using the trapezoidal rule between 0 and 12 h after treatment administration was greater in high- (26 n*M* × h) than low-EL calves (4.30 n*M* × h). Similarly, the maximum concentration of EL in plasma was greater in high- (5.06 n*M*) versus low-EL calves (1.95 n*M*). Furthermore, the time after treatment administration to reach maximum plasma concentration of EL was faster in the high- (4.31 h) compared with the low-EL (4.44 h) treatment. Our results showed that newborn calves were able to absorb EL, suggesting that EL-enriched milk can potentially be used as a natural source of antioxidants to pre-weaned ruminants. We also calculated the apparent efficiency of EL absorption between 0 and 12 h after the oral administration of treatments; calves fed EL-enriched milk tended to have lower apparent efficiency of EL absorption than those fed milk replacer (1.31 vs. 1.80%, respectively). In a study in which 12 healthy volunteers (6 men and 6 women) ingested a single dose of purified SDG (1.31 µmol/kg of body weight), ED and EL reached their maximum plasma concentrations at 14.8 and 19.7 h after intake of SDG, respectively [87]. In addition, the area under the curve of EL (mean = 1762 n*M* × h) increased by 2-fold compared with that of ED (mean = 966 n*M* × h), indicating a greater systemic exposure to EL than ED [87]. Although our study shed some light in the metabolism of milk EL in vivo [58], future research using humans or animal models that better represent the anatomy and physiology of humans’ gastrointestinal tract is warranted to provide further insights about the pharmacokinetics of EL consumed through milk. 

An association between serum EL concentration ≥ 10 n*M* and decreased mortality risk (i.e., all-causes and breast cancer-specific) after breast cancer surgery has been reported in women [88]. Milk concentration of EL averaged 395 n*M* in two studies in which Jersey cows received 15–16% FM [18,19]. Thus, 1 daily serving (250 mL) of EL-enriched milk with a concentration of 395 n*M* of EL would result in 1.3 n*M* of EL in plasma assuming an apparent efficiency of absorption of 1.31% based on our previous work [58]. These results imply that EL-enriched milk needs to be consumed in combination with other lignan-rich foods to reach EL concentration in blood that has been linked to decreased mortality and positive health outcomes in humans [88]. However, our inferences should be interpreted cautiously because calves were fed milk as the sole dietary source [58], which may have increased digesta passage rate ultimately limiting the intestinal absorption of EL. 

## 5. Antioxidant Activity of Flaxseed Products and Dairy Cow Health

Periparturient dairy cows mobilize triacylglycerols from the adipose tissue to support elevated energy demand during early lactation [59,89]. As lactation advances, dairy cows also experience extensive metabolic adaptations for maintenance and high milk production [90]. This increased metabolic activity requires more oxygen consumption, which stimulates production of reactive oxygen species (ROS) [91]. When ROS generation exceeds the endogenous antioxidant defense capacity, animals are susceptible to oxidative damage to DNA, lipids, protein, and other cellular components [92]. Oxidative stress may also impair the immune system of dairy cows [91,93] so that they are likely more vulnerable to a variety of metabolic disorders, including udder edema, milk fever, retained placenta, mastitis, and reproductive issues [90,91]. It has been shown that newborn calves had greater blood concentration of free radicals than pregnant cows, suggesting that they undergo a more severe oxidative stress [20]. Therefore, mitigation of oxidative stress has great potential to improve dairy cattle health and profitability of dairy enterprises. In recent years, several studies were conducted to investigate the effects of flaxseed products on the activity of antioxidant enzymes in plasma and erythrocytes, and their gene expression in mammary and hepatic tissues and results are discussed below. 

Superoxide dismutase, CAT, and GP_X_ are antioxidant enzymes commonly involved in combating free radicals in animals’ blood and tissues. Superoxide dismutase catalyzes the reaction of highly reactive superoxides to form less reactive peroxides [94]. Peroxides can then be converted to water and oxygen under the catalyzation of CAT [95]. Glutathione peroxidase is an enzyme that facilitates reduction reactions of hydroperoxides such as organic hydroperoxides and peroxides [94]. According to previous work [96], CAT mainly works against free radicals when animals experience severe oxidative stress, whereas GP_X_ protects those with less oxidative stress pressure. 

The activity of antioxidant enzymes in lactating dairy cows fed different flaxseed products are summarized in Table 3. Overall, the activities of SOD, CAT, and GPx in plasma, erythrocytes, and mammary and hepatic tissues were not affected by supplementation of FH, FM, WF, and whole linola (see Table 3). Linola is a cultivar of flaxseed containing approximately 70% linoleic acid [97]. A potential explanation for the inability of flaxseed products to modify the activity of antioxidant enzymes in most studies listed in Table 3 may be due to the use of mid-lactation dairy cows experiencing low oxidative stress. Contrarily, a study [98] reported that inclusion of 12.4% FM lowered plasma CAT activity and tended to elevate that of erythrocytes. Likewise, a tendency for increased activity of SOD in mammary tissues was observed with feeding 9.88% FH [56]. It is also important to note that significant treatment by sampling time interactions were found for plasma CAT and GPx activity with FM supplementation [99]. Plasma CAT and GPx activity responded quadratically and cubically to increasing amounts of FM (0, 5, 10, 15%) when blood samples were collected before feeding, but no treatments effect was observed with sampling 3 h post-feeding [99]. These interactions were probably caused by a longer-lasting supply of antioxidants from the diet with the greatest intake of SDG (i.e., 15% FM) compared with the lower levels [99]. 

The effect of flaxseed products on mRNA abundance of antioxidant enzymes genes in the mammary gland of lactating dairy cows are summarized in Table 4. Feeding 9.88% FH [56] and incremental amounts of FM (0, 5, 10, and 15%) [99] increased mRNA abundance of CAT gene, whereas no changes were observed with inclusion of 13.7% FM [100]. Additionally, GPx1 and GPx3, two isoforms of GPx, were not impacted with feeding varying amounts of FM [98,99,100]. However, GPx1 and GPx3 were up- and downregulated, respectively, in dairy cows fed 9.88% FH compared with those fed the control diet [56]. These contradictory effects on GPx1 and GPx3 mRNA abundance with feeding 9.88% FH may be associated with different functions of GPx genes [101]. In addition to CAT and GPx, the mRNA abundance of three isoforms of SOD genes including SOD1, SOD2, and SOD3 were quantified. Both De Marchi et al. [100] and Schogor et al. [99] showed that the mRNA abundance of SOD genes was not modified by FM supplementation to lactating dairy cows. In contrast, an increase in the mRNA abundance of SOD1 and decreases in that of SOD2 and SOD3 were detected in dairy cows fed 9.88% FH [56]. The promoter region of SOD1 contains an antioxidant response element not found in SOD2 and SOD3, thereby consistent with the variable responses of SOD genes to FH supplementation [102]. Collectively, the effects of flaxseed products on modifying antioxidant enzymes or their expression in mammary or hepatic tissues were limited.

The nuclear factor (erythroid-derived 2)-like 2 (*NFE2L2*) relative mRNA abundance in mammary tissues increased linearly in cows fed incremental amounts of FM [99] (see Table 4). The *NFE2L2* gene encodes for a transcription factor involved in activating the expression of a series of genes that are transcribed and translated into antioxidant proteins [103,104]. It is noteworthy that increased *NFE2L2* [99] did not coincide with changes in mRNA abundance of most antioxidant enzymes as discussed above. A trend was observed for decreased relative mRNA abundance of the nuclear factor kappa-light-chain-enhancer of activated B cells subunit 1 (NF-κB1) gene with feeding 13.7% FM to lactating dairy cows [100]; however, two other studies [98,99] did not detect changes in mRNA abundances of NF-κB and NF-κB1, respectively, when similar amounts of FM were fed. The NF-κB1 gene is one of the five members of the NF-κB family, which regulates numerous genes involved in inflammatory and immune responses, apoptosis, and tumor progression [105,106,107]. The polyphenolic compound quercetin protected interstitial Leydig cells against atrazine-induced toxicity by decreasing the expression of NF-κB and preventing oxidative stress [107]. As shown in Table 2, FM is the richest source of the lignan SDG, a polyphenolic compound like quercetin, thus in line with the reduced expression of NF-κB1 gene [100]. These results suggest that FM supplementation has potential to decrease inflammation and cell death in mammary tissues [100]. Interestingly, decreased NF-κB1 was not associated with changes in the relative mRNA abundance of antioxidant enzymes [100], possibly because FM supplementation did not affect the nuclear factor erythroid 2–related factor 2 (*NRF2*) mRNA abundance, which agrees with previous work [98]. As known, *NRF2* is a transcription factor that activates the expression of multiple genes holding an antioxidant response element in their promoters for codifying antioxidant proteins and phase 2 detoxifying enzymes [105]. Future research is needed to better understand how the relationship between flaxseed supplementation and expression of antioxidant enzyme genes may interact to modulate inflammatory, immunological, and health responses in dairy cows experiencing oxidative stress.

Thiobarbituric acid-reactive substances (TBARS) are markers of oxidative status and mainly used to estimate oxidative damage to lipids or lipoperoxidation [109]. Lipoperoxidation can cause damages to cell membranes and membrane-bound enzymes [110]. The impact of flaxseed products supplementation on TBARS concentration in milk, plasma, and ruminal fluid are summarized in Table 5. Quadratic and cubic responses for milk TBARS production were observed in cows fed incremental amounts of FM, with 5% FM and 10% FM resulting in the lowest values [99]. They [99] also reported a significant treatment × sampling time interaction for ruminal TBARS concentration; a linear decrease in TBARS was found with increasing FM supplementation at 2 h after feeding, but no changes were detected at 0 (pre-feeding), 4, and 6 h post-feeding. It was hypothesized that the defense of FM-lignans against oxidation in the rumen is a time-dependent process, with protection being more effective within the first hours after feeding and weakening over time [99]. However, another study [111] reported no changes in ruminal TBARS concentration at 0 (pre-feeding) and 2 h post-feeding but decreased thereafter (4 and 6 h) with feeding 12.4% FM. A third experiment [112] showed a significant decrease in ruminal TBARS concentration in dairy cows fed 13.7% FM despite no treatment × sampling time interaction effect. None of the studies listed in Table 5 (i.e., [99,111,112]) reported effects of FM on plasma TBARS concentration. Similarly, no effects of FM supplementation were observed for the plasma peroxidizability index and total antioxidant capacity [111,112]. As pointed out earlier, research using dairy cattle during stages of life (e.g., transition period, neonatal phase, weaning) more conducive of oxidative stress is needed to better assess the role of flaxseed lignans on animal oxidative status and overall health. 

## 6. Conclusions

Our review showed that flaxseed products, particularly FM and FH were effective in enhancing the concentration of EL in milk. The metabolism of SDG to EL by the ruminal microbiota possibly involves deglycosylation, demethylation, dehydroxylation, and dehydrogenation reactions. In vitro work showed that ruminal bacteria from the genus *Prevotella* were the most efficient converters of SDG to SECO. The large interindividual variation in milk EL yield suggests that the ruminal microbiota vary in their effectiveness for metabolizing SDG to EL. This opens the possibility for controlling, through genetic selection or other management tools, the amount of EL consumed by the population. Scientific information related to the pharmacokinetics of EL consumed via milk is limited. Recent research showed that EL is absorbed by newborn dairy calves, indicating that EL-enriched milk has potential to be used as a natural source of antioxidants for pre-weaned ruminants.

We specifically call for future research to assess how the relationship between flaxseed supplementation and expression of antioxidant enzyme genes may interact to modulate inflammatory, immunological, and health responses in dairy cattle experiencing oxidative stress. Microbiome work is also needed to elucidate the profile and function of the ruminal microbiota species and genomes involved in the metabolism of lignans in ruminants. The impact of forage types (e.g., low- vs. high-lignan legumes), forage conservation methods, and different sources of NSC and fibrous by-products (e.g., soyhulls, beet pulp, citrus pulp) on ruminal microbiome and milk EL concentration in cows fed FM deserves specific attention. These complex research questions should be tackled through collaborative efforts of laboratories with complementary expertise so that an in-depth understanding of the opportunities and challenges of lignans research in dairy cattle can be successfully implemented. To do so, the scientific community, dairy processors, and the population need to be informed and engaged concerning the implications of phytoestrogens to animals and humans’ health.

## Figures and Tables

**Figure 1 molecules-24-00041-f001:**
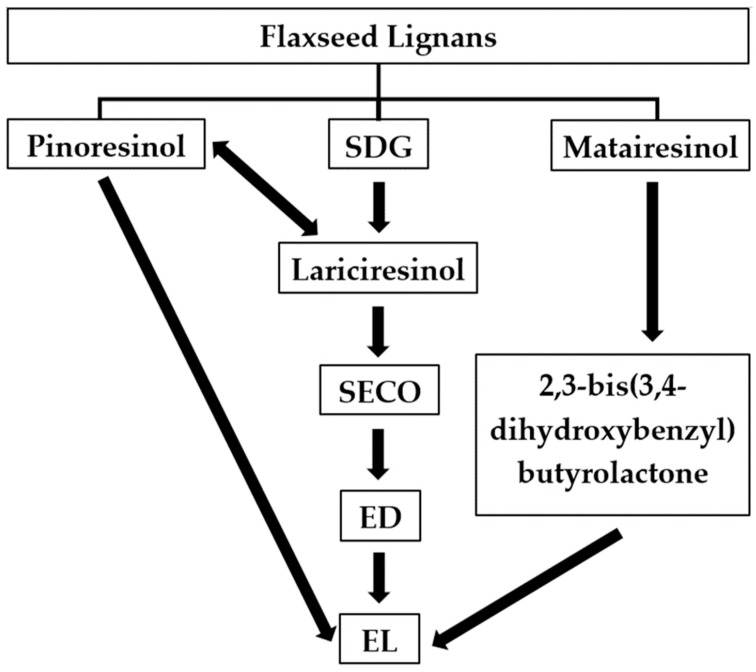
Metabolic pathways for enterolignans production from flaxseed lignans by the gut microbiota of humans. SDG = secoisolariciresinol diglucoside; SECO = secoisolariciresinol; ED = enterodiol; EL = enterolactone. Adapted from [29].

**Figure 2 molecules-24-00041-f002:**
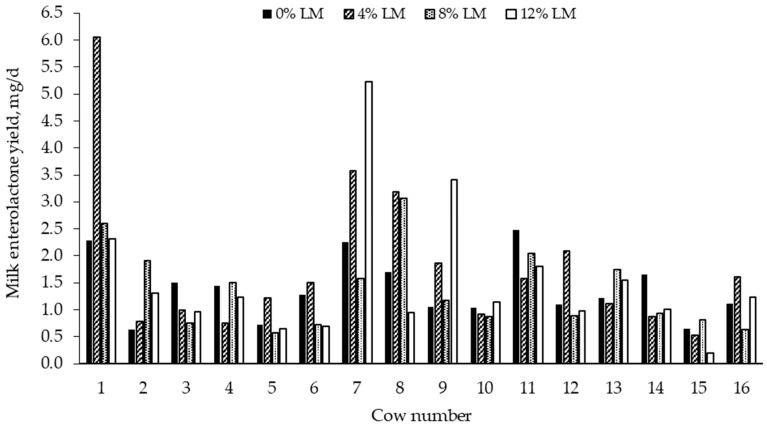
Interindividual variation in milk enterolactone yield in dairy cows fed (% of diet dry matter) diets in which ground corn was replaced by incremental amounts of liquid molasses (LM) (see [18] for study details).

**Table 1 molecules-24-00041-t001:** Milk enterolactone concentration in dairy cows fed different flaxseed products.

References	No. of Cows	DIM ^1^	Experimental Design ^2^	Treatments ^3^	Milk Enterolactone Concentration ^4^
[16]	24	119	RCB	CON, 10% FM, 10% WF	10% FM = 10% WF > CON
[8]	4	92	4 × 4 Latin square	FO & FH at ABO/ABO, RUM/ABO, RUM/RUM, ABO/RUM	ABO/RUM = RUM/RUM > RUM/ABO = ABO/ABO
[11]	32	231	RCB	0%, 5%, 10%, 15% FM	Linear increase
[54]	12	61	RCB	CON, 20% FM	20% FM > CON
[53]	32	175	RCB	0%, 5%, 10%, 15% WF	Tendency for linear increase
[55]	4	190	4 × 4 Latin square	CON, 20% FH, MON, 20% FH + MON	20% FH = 20% FH + MON > CON = MON
[17]	45	140	RCB	0%, 5%, 10%, 15%, 20% FH	Linear increase
[56]	8	163	4 × 4 Latin square	CON, 9.88% FH, 500 g/d FO at ABO, 9.88% FH + 500 g/d FO at ABO	9.88% FH = 9.88% FH + 500 g/d > CON > 500 g/d FO
[48]	6	95	6 × 6 Latin square	2 × 3 factorial: FH (0%, 15.9%) × FO (0, 250, 500 g/d)	15.9% FH diets > 0% FH diets
[18]	16	135	4 × 4 Latin square	2 × 2 factorial: GRC + 16% SBM-SFM mix, GRC + 16% FM, LM + 16% SBM-SFM mix, LM + 16% FM	16% FM diets > 16% SBM-SFM mix diets & LM diets > GRC diets
[47]	8	56	4 × 4 Latin square	2 × 2 factorial: CON, 13.7% FM, 250 g/d SO at ABO, 13.7% FM + 250 g/d SO at ABO	No treatment differences
[57]	8	108	4 × 4 Latin square	2 × 2 factorial: CON, 12.4% FM, 250 g/d FO at ABO, 12.4% FM + 250 g/d FO at ABO	12.4% FM = 12.4% + 250 g/d FO > CON = 250 g/d FO
[10]	8	112	4 × 4 Latin square	0%, 5%, 10%, 15% FM	Linear increase
[58]	16	95	4 × 4 Latin square	CON, 15% FM + 5% sucrose, 15% FM + 3% FO, 15% FM + 5% sucrose + 3% FO	15% FM + 5% sucrose = 15% FM + 3% FO = 15% FM + 5% sucrose + 3% FO > CON
[19]	16	101	4 × 4 Latin square	Different GRC to LM ratios (12:0, 8:4, 4:8, and 0:12) + 15% FM	Tendency for cubic effect

^1^ DIM = days in milk; ^2^ RCB = randomized complete block design; ^3^ CON = control, FM = flaxseed meal, WF = whole flaxseed, FO = flaxseed oil, FH = flaxseed hulls, ABO = abomasum, RUM = rumen, MON = monensin, GRC = ground corn, LM = liquid molasses, SBM = soybean meal, SFM = sunflower meal, SO = sunflower oil; ^4^ Significant differences in the cited references were declared at *p* ≤ 0.05 and trends at 0.05 < *p* ≤ 0.10; no treatment differences (*p* > 0.10).

**Table 2 molecules-24-00041-t002:** Nutritional profile (% of dry matter) of flaxseed products used in studies listed in Table 1
^1^.

Item	Flax Products		
	Flaxseed Hulls ^2^ (*n* = 5)	Flaxseed Meal ^3^ (*n* = 6)	Whole Flaxseed ^4^ (*n* = 1)
Crude protein	22.4 ± 2.41	37.2 ± 1.35	23.5
Neutral detergent fiber	20.6 ± 2.64	30.6 ± 4.61	20.7
Acid detergent fiber	15.8 ± 3.44	15.9 ± 1.39	13.7
Crude fat	28.4 ± 3.09	3.70 ± 4.11	34.9
SDG	1.00 ± 0.08	1.60 ± 0.21	0.60

^1^ Values are presented as mean ± standard deviation, unless otherwise noted. ^2^ Values were calculated using data reported by [8,17,48,55,56]; 4 studies including [8,48,55,56] reported the same nutritional composition for flaxseed hulls except for a different secoisolariciresinol diglucoside (SDG) concentration value reported by [55]; no SDG concentration for flaxseed hulls was reported by [17]. ^3^ Values were calculated using data from [11,18,19,47,57,58]; SDG concentrations were not reported by [47] and [57]. ^4^ Values were calculated using data from [53].

**Table 3 molecules-24-00041-t003:** Activity of antioxidant enzymes in plasma, erythrocytes, and mammary and hepatic tissues in lactating dairy cows fed different flaxseed products ^1^.

Item ^3^	Treatments and References ^2^					
	Non-FH vs. 9.88% FH Diets [56]	0%, 5%, 10%, 15% FM [99]	Non-FM vs. 16% FM Diets [18]	Non-FM vs. 13.7% FM Diets [100]	Non-FM vs. 12.4% FM Diets [98]	CON vs. 7.7% WF, 7.7% WL [108]
Plasma ^4^						
CAT	NS	NS ^8^	−	NS	↓	−
GP_X_	NS	NS ^9^	NS	NS	NS	−
SOD	NS	NS	NS	NS	NS	−
Erythrocytes ^5^						
CAT	NS	NS	−	NS	↑, †	−
GP_X_	NS	NS	−	NS	NS	−
SOD	NS	NS	−	NS	NS	−
Mammary tissue ^6^						
CAT	NS	NS	−	NS	NS	−
GP_X_	NS	NS	−	NS	NS	−
SOD	↑, †	NS	−	NS	NS	−
Hepatic tissue ^7^						
CAT	−	−	−	−	−	NS
GP_X_	−	−	−	−	−	NS
SOD	−	−	−	−	−	NS

^1^ Significant differences in the cited references were declared at *p* ≤ 0.05 and trends at 0.05 < *p* ≤ 0.10; NS = not significant (*p* > 0.10). ^2^ FH = flaxseed hulls; CON = control; FM = flaxseed meal; WF = whole flaxseed; WL = whole linola (linola is a cultivar of flaxseed containing approximately 70% linoleic acid [97]). ^3^ CAT = catalase; GPx = glutathione peroxidase; SOD = superoxide dismutase; ↑ = positive effect; ↓ = negative effect; † = tendency for significance; − = no measurement. ^4^ CAT and GPx units were reported as µmol/min per mg of protein, nmol/min per mg of protein, or nmol/min per mL; SOD units were reported as U/mg of protein, U/g of protein, nmol/min per mg of protein. ^5^ CAT units were reported as µmol/min per mg of protein or nmol/min per g of protein; GPx units were reported as nmol/min per g of protein or nmol/min per mg of protein; SOD units were reported as U/g of protein, U/mg of protein, or µmol/min per mg of protein. ^6^ CAT units were reported as µmol/min per mg of protein, nmol/min per g of protein, or nmol/min per mg of protein; GPx units were reported as nmol/min per g of protein or nmol/min per mg of protein; SOD units were reported as U/g of protein, U/mg of protein, or µmol/min per mg of protein. ^7^ CAT, GPx, and SOD units were reported as U/mg of protein. ^8^ no overall treatment effect, but a significant treatment by sampling time interaction was reported [quadratic and cubic effects before feeding (0 h) and no effect at 3 h post- feeding]. ^9^ no overall treatment effect, but a significant treatment by sampling time interaction was observed [quadratic and cubic effects before feeding (0 h) and no effect at 3 h post- feeding].

**Table 4 molecules-24-00041-t004:** Relative mRNA abundance of oxidative stress-related genes in mammary tissues of lactating dairy cows fed flaxseed products ^1^.

Items ^3^	Treatments ^2^ and References			
	Non-FH vs. 9.88% FH diets [56]	0%, 5%, 10%, 15% FM [99]	Non-FM vs. 13.7% FM diets [100]	Non-FM vs. 12.4% FM diets [98]
CAT	↑	linear increase†	NS	−
GP_X1_	↑	NS	NS	NS
GP_X3_	↓	NS	NS	−
SOD1	↑	NS	NS	−
SOD2	↓	NS	NS	−
SOD3	↓	NS	NS	−
*NFE2L2*	−	linear increase	−	−
NF-κB	−	NS	−	−
NF-κB1	−	−	↓, †	NS
NRF2	−	−	NS	NS

^1^ Significant differences in the cited references were declared at *p* ≤ 0.05 and trends at 0.05 < *p* ≤ 0.10; NS = not significant (*p* > 0.10). ^2^ FH = flaxseed hulls; FM = flaxseed meal. ^3^ CAT = catalase; GP_X_ = glutathione peroxidase; SOD = superoxide dismutase; *NFE2L2* = nuclear factor (erythroid-derived 2)-like 2; NF-κB1 = nuclear factor Kappa-B1; NRF2 = nuclear factor (erythroid-derived 2)-like 2; ↑ = positive effect; ↓ = negative effect; † = tendency for significance; − = no measurement.

**Table 5 molecules-24-00041-t005:** Indicators of oxidative stress in lactating dairy cows fed flaxseed products ^1^.

Items ^3^	Treatments ^2^ and References		
	0%, 5%, 10%, 15% FM [99]	Non-FM vs. 12.4% FM Diets [111]	Non-FM vs. 13.7% FM Diets [112]
Milk TBARS	Q,C ^4^	NS	NS
Plasma TBARS	NS	NS	NS
Ruminal TBARS	NS ^5^	NS ^6^	↑ ^7^
Plasma peroxidizability index	−	NS	NS
Plasma total antioxidant capacity	−	NS	NS

^1^ Significant differences in the cited references were declared at *p* ≤ 0.05 and trends at 0.05 < *p* ≤ 0.10; NS = not significant (*p* > 0.10). ^2^ FM = flaxseed meal. ^3^ TBARS = thiobarbituric acid-reactive substances (nmol of malondialdehyde equivalent/mL); plasma peroxidizability index = (% dienoic fatty acid × 1) + (% trienoic fatty acid × 2) + (% tetraenoic fatty acid × 3) + (% pentaenoic fatty acid × 4) + (% hexaenoic fatty acid × 5) [113]; plasma total antioxidant capacity expressed in m*M*. ^4^ Quadratic and cubic effects were observed. ^5^ no overall treatment effect, but a significant treatment by sampling time interaction was reported [linear decrease at 2 h post-feeding; no changes at 0 (pre-feeding), 4, and 6 h post-feeding]. ^6^ no overall treatment effect, but a significant treatment by sampling time interaction was reported [no effects at 0 (pre-feeding) and 2 h post-feeding but decreased with FM at 4 and 6 h post-feeding]. ^7^↑ = positive effect.

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
