# Peer review of "A Review of Lignan Metabolism, Milk Enterolactone Concentration, and Antioxidant Status of Dairy Cows Fed Flaxseed"

_molecules, 2018, doi:10.3390/molecules24010041_

Round 1
Reviewer 1 Report
Recommendations:
· keywords are quite disordered, some words are the same as in the title
· l. 43, it is really possible to increase global consumption of milk by 60%?
· l. 119-142, the part is not well-arranged, the table would be more appropriate
· l. 204-205 to use whole name of microorganisms
· l. 204, fibrisolvens or fibrosolvens?
· l. 289, the citations, to use either name or number of references
· l. 414, susceptibly or susceptibility?
· Table 3 and Table 4, it would be more appropriate to use P (probability) - values of statistical methods (used in selected manuscripts) than only abbreviation NS (not significant); What does mean “the tendency for significance”?
· some sentences should be rewrite (they are too long or difficult to understanding), for example lines 50-52, 106-110, 449-452, 511-513
· there are sometimes typographical errors in the manuscript, for example l. 101 (missing comma), 171 (..), 361 (souses), 454 (tract.is), 484 (leukocytes), 487 (capacity capacity), Table 5 - CON (0%), 5%, …
· the chapter Conclusions is 1) too long, 2) information given in Abstract and Introduction should not be exactly repeated in Conclusion; only for example, l. 10-13 vs. 31-34 vs. 625-628, 3) l. 643-645, 648-652, it should not be in Conclusions, and again, it was already mentioned
· authors should check the references (typographical errors, abbreviation of journal title, ..), for example l. 694, reference No. 12, l. 720, 806, 808, 815, 831, 851, 855, 920, 965
Author Response
Reviewer 1
Recommendations:
keywords are quite disordered, some words are the same as in the title
AU: The keywords have been reordered and changed to not repeat words in the title of the manuscript. Please see L26-27.
Line 43, it is really possible to increase global consumption of milk by 60%?
AU: We are using projections from FAO. We agree that it seems high, but we are using FAO projections. The text was slightly modified with the correct range (please see L44)
Lines 119-142, the part is not well-arranged, the table would be more appropriate
AU: We agreed that the text was too dense and confusing. We reworded the text to improve clarity (please see L118-145). Most of the studies cited are already summarized in Table 1.
Lines 204-205 to use whole name of microorganisms
AU: Complete species names of ruminal bacteria have been now added at L189-191.
Line 204, fibrisolvens or fibrosolvens?
AU: It seems that both have been used interchangeably. However, we are more familiar with fibrisolvens. Please see L189.
Line 289, the citations, to use either name or number of references
AU: The names of the authors have been omitted throughout the manuscript.
Line 414, susceptibly or susceptibility?
AU: It should be susceptibility. The word “susceptibly” has been replaced with “susceptibility”. Please see L394.
Table 3 and Table 4, it would be more appropriate to use P (probability) - values of statistical methods (used in selected manuscripts) than only abbreviation NS (not significant); What does mean “the tendency for significance”?
AU: We appreciate your suggestion, but we respectfully disagree. Adding P-values in the body of the table may cause potential readers to mistakenly confuse P-values with actual data of a given variable. So, we decided to include a footnote in Tables 3-5 indicating P-values for significance and trends reported in the references. Please see Tables 3-5.
some sentences should be rewrite (they are too long or difficult to understanding), for example lines 50-52, 106-110, 449-452, 511-513
AU: Thanks for pointing this out. These sentences were reworded to improve clarity. Please see L48-51, L105-109, and L471-473. The sentence (449-452) was deleted from this revised version of the manuscript based on comments from Reviewer 2.
there are sometimes typographical errors in the manuscript, for example l. 101 (missing comma), 171 (..), 361 (souses), 454 (tract.is), 484 (leukocytes), 487 (capacity capacity), Table 5 - CON (0%), 5%, …
AU: We appreciate the reviewer diligence to identify these mistakes. They were all fixed. Please see L101, L344, and L423. We decided to delete the sentence that “leukocytes” and “capacity” were used to streamline the text. “CON” was removed from Table 5.
the chapter Conclusions is 1) too long, 2) information given in Abstract and Introduction should not be exactly repeated in Conclusion; only for example, l. 10-13 vs. 31-34 vs. 625-628, 3) l. 643-645, 648-652, it should not be in Conclusions, and again, it was already mentioned
AU: Thanks for bringing this up. The conclusion section was shortened and reworded (please see L581-603).
authors should check the references (typographical errors, abbreviation of journal title, ..), for example l. 694, reference No. 12, l. 720, 806, 808, 815, 831, 851, 855, 920, 965
AU: These inconsistencies were fixed (please see L634-635, L660-661, L741-743, L744-746, L750-751, L766, L805-807, L825, L886-887, and L926-928).
Reviewer 2 Report
This is an interesting review about enterolactone concentrations in milk and their potential effects in cow and human health. In general, this review could be useful as an update for the field. However, authors should be careful with some overinterpretation of data, contradictions and/or lack of relevant information:
-Lines 48-49: “In general flaxseed hulls (FH) and FM have resulted in greater concentrations of EL in milk than whole flaxseed (WF)”. However, in lines 261-262: “However, no differences in the concentration of milk EL was found between FM and WF” Please, clarify.
-Lines 87-89: “The mammalian lignans ED and EL are absorbed in the small intestine followed by conjugation as glucuronides and sulfates based on in vitro work using human colon epithelial cells…” Absorption in the small intestine should not be concluded from studies with colon epithelial cells.
-Lines 292-293: “…ABCG2 may play a role in the interindividual variation of EL secretion in milk”. Related to this interindividual variation, a higher secretion capacity of enterolactone by cows carrying the Y581S ABCG2 polymorphism was shown by higher milk/plasma ratio in these animals (Otero et al., 2016, Animal. 10(2):238-47). This is very useful information that would improve the manuscript.
-Lines 296-297 and 640-642: “…high intake of phytoestrogens may result in adverse health effects, particularly in critical stages of infant development,…” Not only in infant development. Pregnancy and lactation could also be key stages in this context due to changes in milk and body composition (Guarda et al., 2014, Food Chem Toxicol, 69:69-75) and in hormonal and biochemical parameters (Troina at al., 2012, Food Chem Toxicol, 50:2388-96. This is very useful information that would improve the manuscript.
-Lines 443-446 and lines 645-647: As the authors stated, in these cases comparison between species is not possible and the conclusions are too speculative. Please remove.
-Lines 448-449:“… thus suggesting that EL found in ruminants´milk is likely conjugated with glucuronides and sulfates” This is also too speculative since most of the studies analysing EL in milk just measure only enterolactone (by HPLC) and is not conjugated.
Other minor points:
-Lines 11-12, lines 32-33: “”…including weak estrogenic and cardioprotective activities, as well as antiestrogenic,….” This dual effect should be clarified.
-Lines 31-34 are exactly the same as lines 10-13 of abstract. Please, rewrite.
-Line 290: Miguel et al is reference 56 not 55.
-Lines 376-378: Check writing. “Compared with Holsteins in study 10…..were greater in Holsteins in studies 9 and 66…”
Author Response
Reviewer 2
This is an interesting review about enterolactone concentrations in milk and their potential effects in cow and human health. In general, this review could be useful as an update for the field. However, authors should be careful with some overinterpretation of data, contradictions and/or lack of relevant information:
AU: We would like to thank the reviewer for the thoughtful and helpful feedback. We appreciate the acknowledgment that our review manuscript is intriguing and favorable.
Lines 48-49: “In general flaxseed hulls (FH) and FM have resulted in greater concentrations of EL in milk than whole flaxseed (WF)”. However, in lines 261-262: “However, no differences in the concentration of milk EL was found between FM and WF” Please, clarify.
AU: We apologize for this inconsistency. The sentence was deleted.
Lines 87-89: “The mammalian lignans ED and EL are absorbed in the small intestine followed by conjugation as glucuronides and sulfates based on in vitro work using human colon epithelial cells…” Absorption in the small intestine should not be concluded from studies with colon epithelial cells.
AU: Thanks for catching this. We reworded and split the sentence in two. (please see L85-87 and L89-91)
Lines 292-293: “…ABCG2 may play a role in the interindividual variation of EL secretion in milk”. Related to this interindividual variation, a higher secretion capacity of enterolactone by cows carrying the Y581S ABCG2 polymorphism was shown by higher milk/plasma ratio in these animals (Otero et al., 2016, Animal. 10(2):238-47). This is very useful information that would improve the manuscript.
AU: Excellent suggestion. Please see revised text (L273-279).
Lines 296-297 and 640-642: “…high intake of phytoestrogens may result in adverse health effects, particularly in critical stages of infant development,…” Not only in infant development. Pregnancy and lactation could also be key stages in this context due to changes in milk and body composition (Guarda et al., 2014, Food Chem Toxicol, 69:69-75) and in hormonal and biochemical parameters (Troina at al., 2012, Food Chem Toxicol, 50:2388-96. This is very useful information that would improve the manuscript.
AU: Thanks for suggesting these references. However, the work for Guarda et al. (2014) was conducted with flax oil, which is known to not be a source of lignans. We included the reference of Troina et al. (21012) who worked with SDG in rats (please see L284). We also included a statement from a literature review about the benefits and risks of phytoestrogens intake. Please see changed text (L286-291). We reworded the conclusion to keep the text more generalized (please see L586-588).
Lines 443-446 and lines 645-647: As the authors stated, in these cases comparison between species is not possible and the conclusions are too speculative. Please remove.
AU: Agreed. These sentences were deleted.
Lines 448-449:“… thus suggesting that EL found in ruminants´milk is likely conjugated with glucuronides and sulfates” This is also too speculative since most of the studies analysing EL in milk just measure only enterolactone (by HPLC) and is not conjugated.
AU: This sentence was deleted per suggestion and we modified the text (please see L421-424).
Other minor points:
Lines 11-12, lines 32-33: “”…including weak estrogenic and cardioprotective activities, as well as antiestrogenic,….” This dual effect should be clarified.
AU: This appears to be related to different functions of ED and EL on activation of estrogen receptors. Please see revised text (L32-34).
Lines 31-34 are exactly the same as lines 10-13 of abstract. Please, rewrite.
AU: We reworded the abstract (please see L10-11).
Line 290: Miguel et al is reference 56 not 55.
AU: We dropped a couple of reference and added some more. Thus, the reference numbering has changed throughout the manuscript.
Lines 376-378: Check writing. “Compared with Holsteins in study 10…..were greater in Holsteins in studies 9 and 66…”
AU: The sentence was reworded to improve clarity (please see L360-361).
Round 2
Reviewer 1 Report
Comma, l. 424
Reviewer 2 Report
Authors have adequately answered the questions raised by the reviewer.